# Artifactual pyrosequencing reads in multiple-displacement-amplified sediment metagenomes from the Red Sea

Yong Wang[1], On On Lee[1], Jiang Ke Yang[1], Tie Gang Li[2] and Pei Yuan Qian[1]

[1] Division of Life Science, Hong Kong University of Science and Technology, Hong Kong SAR, China
[2] Institute of Oceanography, Chinese Academy of Science, Qingdao, China

## ABSTRACT

The Multiple Displacement Amplification (MDA) protocol is reported to introduce different artifacts into DNA samples with impurities. In this study, we report an artifactual effect of MDA with sediment DNA samples from a deep-sea brine basin in the Red Sea. In the metagenomes, we showed the presence of abundant artifactual 454 pyrosequencing reads over sizes of 50 to 220 bp. Gene fragments translocated from neighboring gene regions were identified in these reads. Occasionally, the translocation occurred between the gene fragments from different species. Reads containing these gene fragments could form a strong stem-loop structure. More than 60% of the artifactual reads could fit the structural models. MDA amplification is probably responsible for the massive generation of the artifactual reads with the secondary structure in the metagenomes. Possible sources of the translocations and structures are discussed.

## INTRODUCTION

The development of pyrosequencing techniques has brought unprecedented opportunities to environmental microbiological studies (*Logares et al., 2012*). Microbial metagenomes from a variety of ecological settings have been obtained and microbial communities in unique habitats are increasingly uncovered by bar-coded pyrosequencing of 16S ribosomal RNA amplicons (*Biddle et al., 2008*; *Ferrer et al., 2012*; *Huse et al., 2008*). We are able to determine the composition of microbial communities and their roles in elemental cycles by the analyses of pyrosequencing data. Novel genes and pathways involved in new metabolisms and adaptation mechanisms can be predicted and validated in subsequent experiments (*Singh et al., 2009*). As a result, microbial ecology has rapidly developed in recent years. However, the quality of pyrosequencing on different platforms is still a major concern (*Quail et al., 2012*). For example, the ROCHE 454 platform shows weakness in deciphering homopolymers, which account for about 40% of its sequencing errors (*Huse et al., 2007*). Artifactual duplications represent 11–35% of the raw reads generated by the 454 platform (*Gomez-Alvarez, Teal & Schmidt, 2009*). Moreover, some DNA samples from

Corresponding author
Pei Yuan Qian, boqianpy@ust.hk

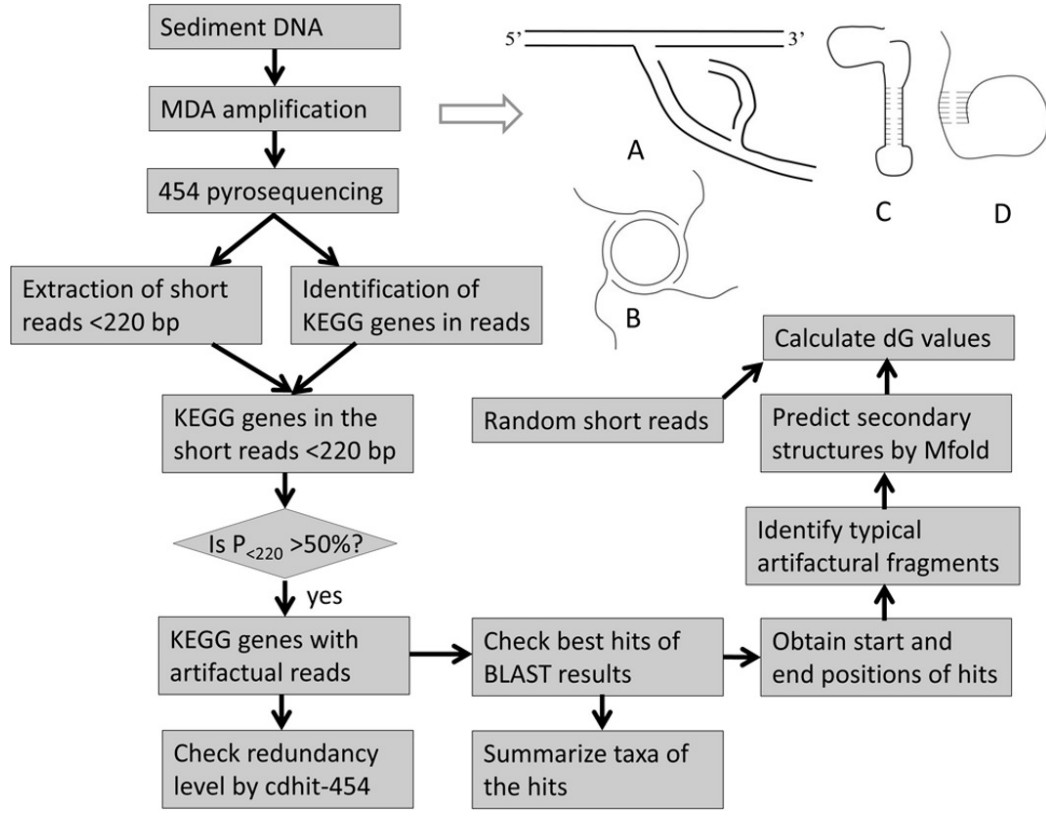

**Figure 1** MDA protocol and flow chart of experiment. (A) The normal MDA reaction on the DNA template; (B) The plasmid is amplified by MDA; (C–D) Two DNA fragments with a complex secondary structure to be amplified by MDA in an unknown manner.

extreme environments need to be amplified by Whole Genome Amplification (WGA) to meet the minimum requirement for the pyrosequencing (*Dean et al., 2002*). Enough DNA sample from the bacterial strain of interest in an environmental sample can then be subjected to pyrosequencing, which enables direct assessment of genomes of individual bacteria, and bypasses the isolation and cultivation procedure in the laboratory.

Despite technical improvements, WGA still has many problems in the amplification of small amounts of DNA. As one of the widely used WGA protocols, MDA uses the phi29 DNA polymerase and random primers to amplify DNA templates (*Dean et al., 2002*; *Dean et al., 2001*). The typical process of MDA amplification is illustrated (Fig. 1A). It has been successfully used to amplify DNA samples from different small biological specimens (*Lasken & Stockwell, 2007*; *Raghunathan et al., 2005*; *Zhang et al., 2006*). But MDA also introduces problems into the amplified DNA sample. Firstly, amplification bias and errors cannot be avoided. Secondly, undesired background amplification may occur and occasionally occupy about 70% of the final MDA product (*Raghunathan et al., 2005*). Therefore, small exogenous DNA contamination and plasmids (amplified as shown Fig. 1B) as the major sources of error should be removed from the DNA template (*Zhang et al., 2006*). Another source of background amplification is

template-independent, primer-primer amplification, accounting for up to 75% of the total yield (*Spits et al., 2006*). It is intensified by a low concentration of DNA template and exogenous DNA contamination (*Pan et al., 2008*). This problem has however been recently resolved by using constrained-randomized primers that cannot hybridize with each other (*Zhang et al., 2006*). Thirdly, chimeras and translocations were frequently identified in MDA amplified samples (*Lasken & Stockwell, 2007*; *Zhang et al., 2006*). A report showed that hundreds of chimeras with DNA rearrangements were identified in 454 reads for an *Escherichia coli* genome from a single cell after MDA amplification (*Lasken & Stockwell, 2007*). Most of them have a sequence inversion that allows the formation of inverted repeats. The occurrence of chimeric sequences was regarded as a result of the incorrect interaction between nearby concurrently synthesized sequences (*Lasken & Stockwell, 2007*). Although these chimeras can be identified and filtered later, this finding is a reminder of other unknown problems during the MDA process. Before we can resolve the technical issues completely, conclusions based on the metagenomic analysis must be treated cautiously. Therefore, there is an urgent need to learn about all the weaknesses in sample treatment protocols and pyrosequencing platforms.

Generally, coastal sediments are rich in microbes and therefore a DNA sample extracted with traditional methods is sufficient for pyrosequencing. However, in deep-sea sediments, the bacterial biomass is low due to harsh environments, which necessitates the use of MDA for metagenomic studies in these extreme biospheres. Raw DNA samples extracted from sediments often contain extracellular DNA and plasmids (*Pietramellara et al., 2009*). The former arises from the lysis of dead cells (*Levy-Booth et al., 2007*). The presence of the non-genomic DNA will raise the background amplification during MDA amplification. In this study, microbes from a deep-sea saline basin in the Red Sea were studied. Although DNA had been extracted, the amount was not large enough for 454 pyrosequencing. MDA amplification had to be used to amplify the DNA samples. However, in this sediment, extracellular DNA was probably abundant because it can be preserved in the saline anaerobic environment (*Borin et al., 2008*). On the other hand, the extracellular DNA samples may have stable secondary structures (Figs. 1C–1D) to resist degradation naturally (*Steinberger & Holden, 2005*). The presence of the contaminant in our samples can be used to examine the biasing effects of MDA amplification. We pyrosequenced MDA-amplified DNA samples from five subsuperficial layers in a sediment core. The assessment of the biasing effects can be determined by examining over-abundant genes in pyrosequenced metagenomes. Several genes with abundant short reads in the metagenomes from the deep layers were studied. These reads generally contained two gene regions (gene fragments). Translocations of the gene fragments were identified in the reads and stem-loop structures could be constructed by the translocated subsections, indicating that multiplication of the fragments was probably triggered by the secondary structure. Hence we conclude that the observed abundant short reads are artifacts of the MDA treatment.

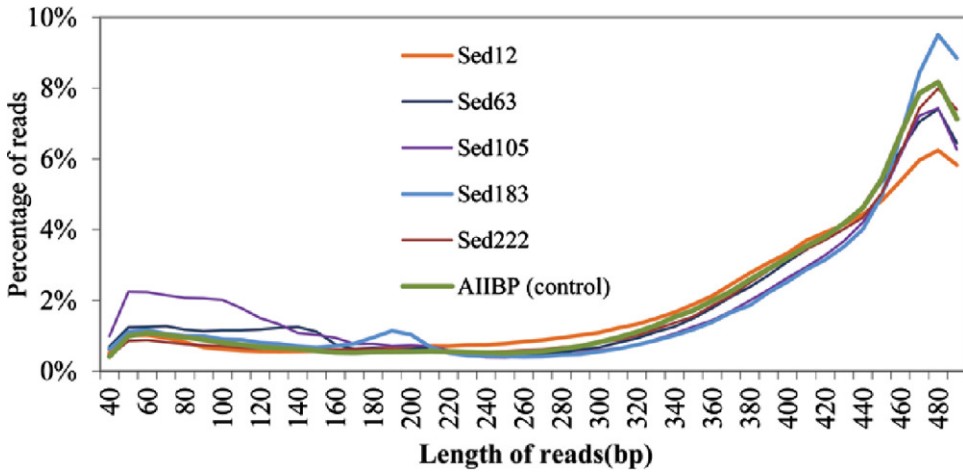

**Figure 2** Length range of the reads for five sediment samples. The control was the metagenome from the overlying Atlantis II brine water.

## MATERIALS AND METHODS

A 2.25-m gravity sediment core was obtained from the Atlantis II Deep (21°20.76' N, 38°04.68' E) in the Red Sea in 2008 (*Swift, Bower & Schmitt, 2012*). Sediment slices of 12–15 cm (Sed12), 63–66 cm (Sed63), 105–108 cm (Sed105), 183–186 cm (Sed183), and 222–225 cm (Sed222) were used for DNA extraction. Ten grams of sediment from the five layers were used for DNA extraction. The crude DNA was purified with an MO BIO Power Max soil DNA isolation kit (Solana Beach, CA, USA). A REPLI-g MDA kit (Qiagen, Hilden, Germany) was applied to amplify the microbial genomic DNA from the sediment layers, followed by pyrosequencing on a ROCHE 454 FLX Titanium platform.

A flowchart of data analysis is illustrated in Fig. 1. A protein database was downloaded from the Kyoto Encyclopedia of Genes and Genomes (KEGG, http://www.genome.jp/kegg, v51). Pyrosequencing reads were used to BLASTX (BLAST2.2.20) against the protein database, with parameters of "-p blastx −e 0.0001 −m 8 −Q 11". Reads for the same KEGG genes were pooled and then sorted into different length ranges in a size increment of 10 bp. The percentage of reads in each of the ranges was calculated. The position of the reads aligned on the full length proteins was determined by the above BLASTX results. If the proteins belonged to the same genus, the protein sizes were generally the same. In each section (10 aa) of the protein, the number of the aligned reads was recorded and the percentage of the reads in all those for the gene was calculated. If the reads were derived from more than one genus according to the result of the best BLAST hits, the proteins in the best hits were first subjected to multiple alignment by ClustalW (www.clustal.org) and then the unaligned parts from both ends were trimmed away. The matching positions of the reads on the proteins were then adjusted to those of the trimmed proteins.

After the reads were sorted into different KEGG genes, short ($<220$ bp; see the distribution shown in Fig. 2) and long reads ($>220$ bp) were separated and counted as $N_{<220}$ and $N_{>220}$. The percentage of the short reads ($P_{<220}$) in a gene was then calculated. An interesting observation was that a majority of these short reads did not

have a full-length alignment with a reference protein by the BLAST search. This means that a small part of them could not be matched to known genes under the current searching criteria. Start and stop points in the alignments were then recorded. After hotspots of the alignment start and stop positions were revealed, the flanking parts (>2 bp) were split out and converted to the sequences on the same strand. Gene fragments in these flanking parts were searched again using the BLASTX program with the default settings, which were more relaxed than those used in the previous searching. Both 5' and 3' flanking sequences were then aligned by MUSCLE v3.6, separately (*Edgar, 2004*).

The short reads were sorted into groups with respect to their alignment positions relative to the hotspots on the proteins. DNA secondary structures of representative short reads were constructed using the Mfold web server (*Zuker, 2003*). Default settings for folding temperature, window size and ionic conditions were employed. To calculate the free energy of all the short reads, UNAFold (*Markham & Zuker, 2008*) was used. The average and standard deviation of the free energy values were then calculated. To compare free energy of the short reads with the other reads in the metagenomes, long reads >300 bp were randomly truncated into short reads. Since the length of a DNA sequence is critical to the measurement of free energy, the average length of the short reads and random reads in a pairwise comparison should be similar. The average lengths of the short reads for different genes ranged between 100 and 160 bp, and thus random reads were further selected to generate four groups of random short reads, with an average length of 100, 120, 140 and 160 bp, respectively. Free energies of the short reads and the random sequences were compared by a Mann-Whitney test in a SPSS package (16.0).

The redundancy level of the reads belonging to different genes in Table 1 was checked by cdhit-454 (*Niu et al., 2010*). Similarity of matching parts in the reads was set at 97%, lower than the threshold suggested by the pyrosequencing error rate, and then clusters among the reads were identified. During the check, long reads were retained in each cluster for further removal of redundancy. If at least 50% of a long read was covered by a short read, and if at least 95% of a short read could be aligned on a unique one, the two reads were clustered.

# RESULTS

## Short artifactual pyrosequencing reads

We obtained 922,401, 480,994, 576,444, 489,923, and 1,099,605 raw pyrosequencing reads with an average length of 410, 382, 358, 397 and 402 bp, for Sed12, Sed63, Sed105, Sed183 and Sed222, respectively. Normally, the pyrosequencing platform produced reads in size of about 400 bp. Therefore, at least Sed63 and Sed105 metagenomes contained an unexpectedly high proportion of short reads. The distribution of all the pyrosequencing reads in different length ranges is shown in Fig. 2. A metagenome from the overlying brine water was used as the control to show over-abundant short reads in the sediment metagenomes. The short reads in Sed12 and Sed222 showed a similar distribution pattern to those in the control, whereas abnormally abundant short reads were observed in the other samples, such as those of 100–150 bp in Sed63, of 50–160 bp in Sed105 and of 180–200 bp in Sed183 (Fig. 2). Thus, these reads might contain pyrosequencing artifacts.

**Table 1** Layer-specific overabundance of short reads for some genes. The KEGG genes in the table were abundant in short reads in sizes of <220 bp (the number is shown). The number of all the reads and percentage of the short reads <220 bp are shown for the individual genes. Lengths and dG values are given with +/− standard deviation in the parentheses.

| Sample | KEGG id | No. <220 bp | Total | % $N_{<220}$ | Clusters | Average length | Average dG |
|--------|---------|-------------|-------|--------------|----------|----------------|------------|
| Sed12 | K04567 | 161 | 294 | 55 | 237 | 162(33) | −28.5(8.3) |
| Sed63 | K06988 | 2208 | 2435 | 91 | 648 | 162(35) | −42.7(14.3) |
| | K07115 | 2056 | 2076 | 99 | 493 | 120(29) | −28.0(10.3) |
| | K00859 | 140 | 192 | 73 | 63 | 161(41) | −40.0(14.1) |
| | K02652 | 68 | 90 | 76 | 69 | 147(37) | −38.3(13.5) |
| | K01061 | 60 | 79 | 76 | 63 | 147(37) | −31.7(10.9) |
| Sed105 | K01440 | 2511 | 2528 | 99 | 462 | 109(23) | −28.6(13.1) |
| | K01409 | 566 | 894 | 63 | 493 | 159(37) | −38.5(12.9) |
| | K09800 | 405 | 461 | 88 | 277 | 149(37) | −32.4(13.4) |
| | K01207 | 124 | 221 | 56 | 188 | 143(42) | −24.8(9.2) |
| | K07788 | 68 | 100 | 68 | 85 | 163(40) | −35.4(11.7) |
| Sed183 | K00257 | 662 | 836 | 79 | 442 | 104(27) | −26.3(8.7) |
| | K09705 | 289 | 301 | 96 | 135 | 115(21) | −35.5(10.7) |
| | K00162 | 177 | 190 | 93 | 123 | 112(26) | −28.0(7.8) |
| | K07506 | 121 | 155 | 78 | 109 | 143(27) | −31.1(6.0) |
| Sed222 | K01627 | 6068 | 6180 | 98 | 1189 | 143(38) | −34.4(116.7) |
| | K01589 | 61 | 113 | 54 | 108 | 105(31) | −25.6(11.4) |

To identify the artifacts, the short reads containing gene fragments were particularly focused on. By comparing the gene fragments in the reads with homologous genes, chimeras and rearrangements that possibly occurred during MDA might be revealed. Several genes were over-represented in the short reads. Fragments of K06988 gene (encoding a protein involved in utilization of DNA as a carbon source) and K07115 gene (encoding putative NADP oxidoreductase coenzyme) accounted for 40 and 37% of all the short reads of Sed63, respectively. The short reads in Sed105 were overwhelmed by K01440 gene fragments (encoding nicotinamidase), occupying 40% of its total short reads. About 9% of the short reads in Sed105 were assigned to K01409 gene (encoding O-sialoglycoproteinendopeptidase), the second most abundant gene. Fragments of K01627 gene (encoding KDO 8-P synthase) represented 37% of the short reads in Sed222, although the distribution of Sed222 reads between 50–220 bp appeared to be normal (Fig. 2). For Sed12 and Sed183, none of the genes was as over-represented as those in the three samples mentioned above; yet 15% of the short reads were assigned to K09705 gene (a hypothetical gene) in Sed183 as indicated by the abnormal bulge in 50–160 bp in Fig. 2.

The reads shorter than 220 bp were then counted for all the KEGG genes. With a cutoff of $P_{<220}$ at 50%, 11 more genes had abundant short reads (Table 1). The alignments of the reads with their KEGG homologous genes in Table 1 were deposited in the European Nucleotide Archive (http://www.ebi.ac.uk/ena/; accession: ERP002361). Overall, only one gene, K04567 (encoding lysyl-tRNA synthetase), was found for Sed12, but all the

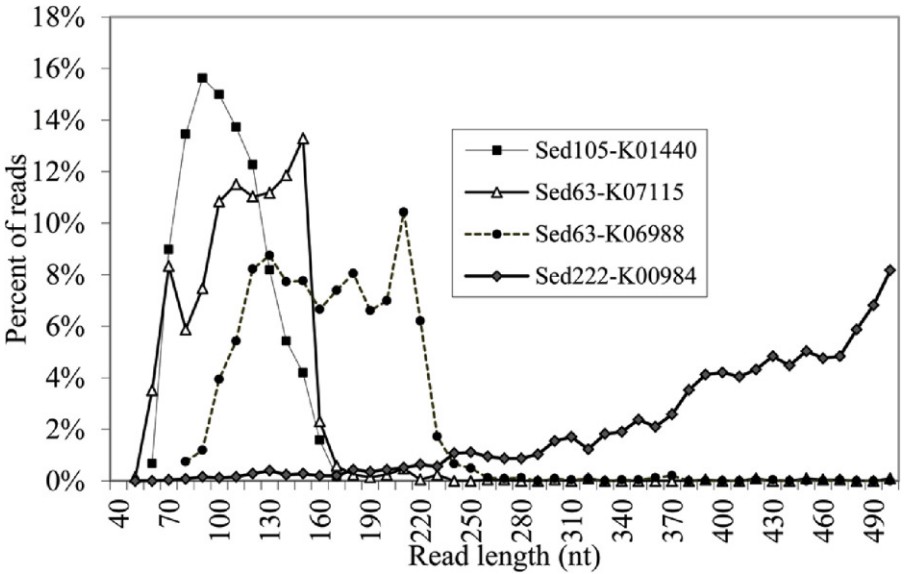

**Figure 3** Length distribution of the reads for the genes with abundant short reads. Alignment positions of the reads on proteins were based on BLASTX results. The numbers in parentheses following the sample names are those of the short reads (<220 bp) and total reads.

reads totaled 294. The genes, K07115, K01440 and K01627, had a remarkably high $P_{<220}$ (>98%). Statistical analysis was performed on the length distribution of these genes by using a reference gene K00984, which had the most abundant reads in all the samples and showed a normal length distribution (Fig. 3). When $N_{<220}$ and $N_{>220}$ were used to form a 2-way contingency table with the corresponding data of K00984, significantly higher numbers of reads were shorter than 220 bp for these genes ($P < 0.0001$). Under this criterion, there were many other genes also showing significant differences. The short reads for these genes were potentially artifactual reads if they were highly similar or concentrated in a small region of the corresponding genes.

Similarity between the reads for the over-represented genes was assessed by arranging them into clusters with a similarity cutoff of 97%. Most of the reads, particularly with a high % $N_{<220}$, were redundant. For example, 6180 reads for K01627 (% $N_{<220} = 0.98$) were sorted into 1189 clusters, and redundancy was up to 80%. The ratio of cluster number to the total was negatively correlated to % $N_{<220}$ ($R = 0.77$) (Table 1). This means that most of the redundancy occurred in the short reads <220 bp. However, compared with other KEGG genes, the number of clusters is still large. This implies the presence of an abundance of sequence variants in the artifactual reads, which may have different sizes and internal polymorphisms.

## Artifactual reads were concentrated on a certain gene region

More details regarding the length distribution of the short reads are provided. Taking the genes K07115, K01440, and K01627 as examples, the sizes of most of the short reads for K01440 in Sed105 ranged narrowly between 70 and 120 bp, whereas the length range of the short reads for K07115 and K06988 in Sed63 was widened to 70–160 and 110–220 bp,

respectively (Fig. 3). The latter two also showed a sharp increase in the frequency of the short reads at ranges of 150–160 and 210–220 bp, respectively. The shortest average size of the short reads was 104 bp in K00257, with a standard deviation of 27 bp; the longest was 163 bp in K07788 (Table 1). The length distribution of the reads for these genes was far from the expected pattern exemplified by the distribution of the reads for K00984 (Fig. 3).

Whether the artifactual reads were located at a certain gene region was examined. The aligned parts derived from BLASTX search were pin-pointed. Hotspots where alignment started and ended were recognized. For example, 117 aa, 131 aa and 186 aa in K06988 protein were the hotspots for the alignments between the artifactual reads and the protein (Fig. S1); 146 aa, 206 aa and 208 aa in K01627 protein were the most frequent points on which the alignment of the artifactual reads started or ended. Table S1 lists more such boundaries. Most of the short reads overlapped in the centre of the protein region and were thus confirmed to be artifacts. For the K06988 protein, the overlapping peaked at 160–170 aa, in which 89, 66, 50 and 56% of the artifactual reads were located in Sed63, Sed105, Sed183, and Sed222, respectively (Fig. S2A). Therefore, the presence of the abundant artifactual reads for K06988 gene was not specific to one sediment layer. The artifactual reads (with $P_{<220} > 50\%$) were found in all the samples except Sed12. There were only seven reads for the homolog in Sed12, and no peak was shown at the peaking range. The gene K00984 was taken as a control again because it did not have abnormally abundant reads in all the samples. The reads for all the homologs were evenly located on the protein and there were no notable abnormal distributions (Fig. S2B).

The artifactual reads aligned to the highly covered K06988 protein region, i.e. 120–190 aa, were examined in more detail. In Sed63, the reads were similar to the homolog Reut_B5423 from *Ralstonia eutropha*. In this region, the similarity of the aligned regions was up to 69% until the end of the region (190 aa). The similarity at the boundary showed a sudden decline to about 50% (Fig. S3). This decline was negatively correlated with the change in read length. There was a gradual decrease of average length of the reads in the protein region (Fig. S3). The highest similarity corresponded with the lowest average read length of 260 bp. In comparison with the other regions, the average length of the reads aligned to this region declined by 45%. The two ends of the alignments between the short reads and the protein encoded by Reut_B5423 were somewhat highly concentrated at the hotspots. We pinpointed 47% of the alignment end positions at 186 aa; 34% of the start points were found at 117 aa and 131 aa (Fig. S1). It was also true for the non-Sed12 samples in spite of their smaller read numbers for K06988. This result further supports the presence of artifactual reads for the Reut_B5423 homolog.

## Stable secondary structures in genes and artifactual reads

The characteristics of the read alignments with the proteins were indicative of special features in the corresponding regions in genes. DNA secondary structures were then examined in the subsections between the alignment hotspots in the artifactual reads. Three typical secondary structures were observed for the artifactual reads for K06988 gene (Fig. 4). At 37°C, a high free energy of folding was observed in the reads with average

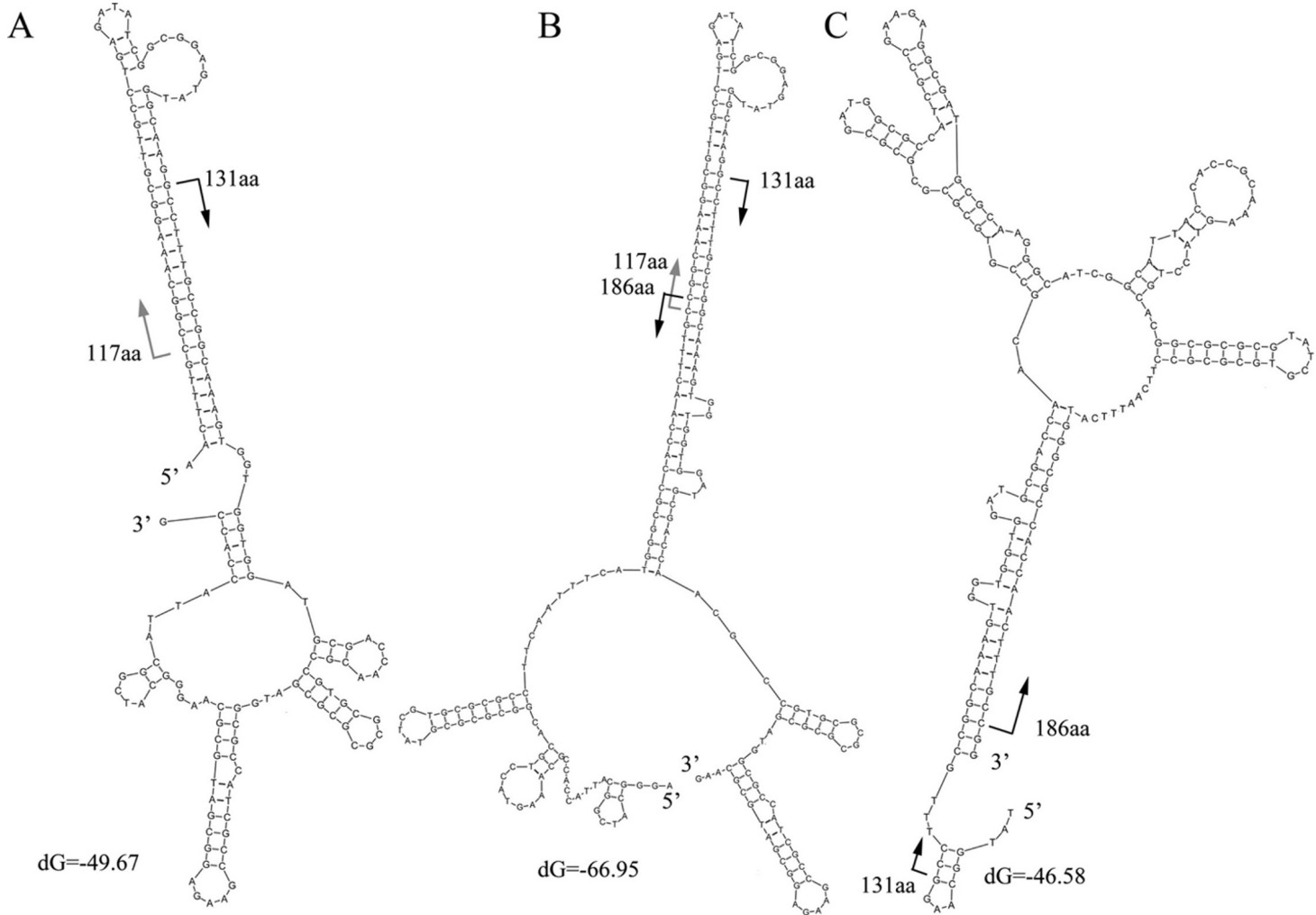

**Figure 4** Secondary structure of three representative reads for K06988 gene. Protein positions of K06988 gene are present on the reads. Length of read A is 152 nt, and 8–151 nt of this read was aligned to 117–164 aa of the K06988 protein. Length of read B is 210 nt and 2–85 nt of this read was aligned to 159–186 aa of the protein; the region of 84–206 nt was aligned to 117–157 aa. Length of read C is 179 nt, and 10–177 nt of this read was aligned to 131–186 aa. The protein positions were indicated by arrows on the reads.

dG equal to −58.1, −39 and −46.1, respectively. Results showed that those started from 117 aa and 131 aa and could fold into a stable secondary structure with a long stem at the 5′ end (Fig. 4A). On the other hand, the structures for those ending at the 186 aa were associated with a long stem at 3′ end (Fig. 4C). However, the stable folding of the reads in Fig. 4A was completely attributable to the introduction of an inserted fragment matched to the region of 117–131 aa. In BLASTX results, the alignment of this fragment between 117 aa and 131 aa of the protein was, in fact, much relaxed. The similarity at the DNA level was merely 53% and almost none at protein level. BLASTN search did not find similar sequences in the NCBI for this fragment in the read. Actually, the part was a reverse complement of the downstream gene region starting from 131 aa (Fig. 4A). It is worthwhile to note that the sequence 5′–TTTGCCGGCAAA-3′ in Fig. 4A was a small inverted repeat

of itself and could introduce more complex structures. An additional folding style was recognized in some reads for K06988. They were two merged gene fragments but with a special arrangement of the fragments (Fig. 4B). The region ending at 186 aa and upstream were translocated upstream of 117 aa position. The translocation resulted in a more stable structure with a dG of −66.95. The upper half of the stem was nearly identical to the 5' stem in Fig. 4A, while the bottom half was the same as the one found at the 3' in Fig. 4C. Likewise, the inserted fragment was along with its downstream sequence, indicating that this unknown fragment had been integrated within the region around 131 aa before the translocation. More variants were observed due to internal slippage regions up to 30 bp.

Moreover, the structure shown in Fig. 4C differs from that for the corresponding gene region (Fig. S4). When the read and the gene were compared, several nucleotide replacements in the read were found to make the secondary structure of the read much more stable. At the 5' end of the read, four replacements were observed; at the 3' end, five replacements were found with four Gs on the modified gene.

The artifactual reads assigned to the other genes were also tested for free energy. Short reads randomly trimmed down from long reads were used for a comparison. Results showed that free energy of the artifactual reads for these genes was significantly lower than that of the random reads of a similar size (U-test; $p < 0.0001$) (Fig. 5). On the contrary, the reads belonging to K00984 had even significantly higher free energy than the random reads (U-test; $p < 0.0001$) (Table S2).

## More translocation cases were frequently observed

The frequency of the translocations occurred in the artifactual reads for K06988 gene was examined. There were about 125 artifactual reads whose secondary structures were clearly shown in Fig. 4B. Possibly, upstream and downstream regions in the models shown in Figs. 4A and 4C also contained the translocated fragments. They could not be recognized, possibly due to the settings of the BLASTX search. A total of 1554 artifactual reads for K06988 gene Reut_B5423 were collected, and the flanking regions (>2 bp) of the alignment hotspots were extracted. Of them, 272, 5' flanking sequences of 131 aa hotspot were recognized as the short relics between the 117 aa and 131 aa. These short flanking sequences with an average length of 12 bp would not form the 5' long stem as shown in Fig. 4A for most of the artifactual reads. In contrast, the 5' flanking sequences of 379 reads with an alignment start position at 117 aa were all matched to the upstream of 186 aa position. The extension to the upstream varied among the sequences and was, on average, 36 bp in size. A total of 937 reads were checked for the genes in their 3' flanking regions of 186 aa position (average length was 48 bp). Alignment of the sequences showed that they were at the downstream region of 117 aa position. This suggests again that the unknown fragment corresponding to the region of 117–131 aa was a natural extension of the 5' of the 131 aa position. Overall, the translocation was detected in at least 60% of all the artifactual reads for the K06988 gene. The reads with alignment positions close to the hotspots were not taken into account, and therefore more such translocation events could be found in other reads. For K01627 genes in Sed222, at least 69% of the 5,921 artifactual

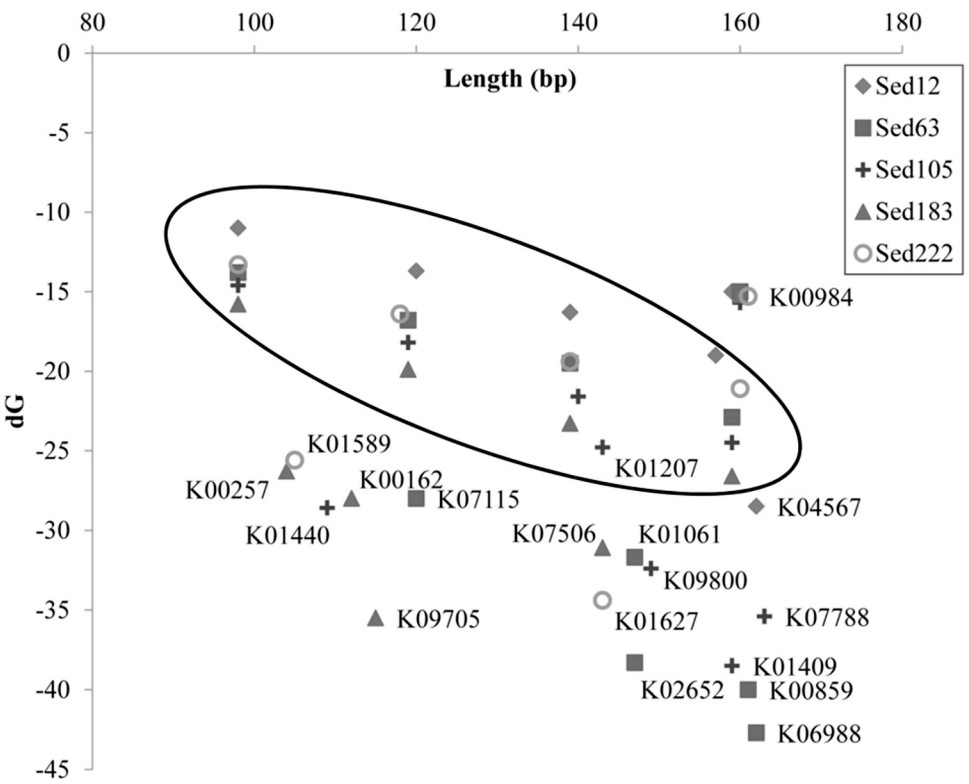

**Figure 5** dG values of randomly-trimmed short reads from the metagenomes and those for selected KEGG genes. The names of the genes are shown beside the symbol of samples in which the average free energy was calculated for their reads. Symbols for the randomly-trimmed short reads in sizes of about 100, 120, 140 and 160 aa do not have a gene name beside them and were circled.

reads contained the translocation fragments. As such, the translocation as shown in Fig. 4B seems quite frequent in the artifactual reads. Difference between them under the stem-loop model was the size of the flanking regions.

## Chimeric gene fragments in short reads

The translocated fragments in the artifactual reads were occasionally derived from different species. Chimeras were thus observed in the reads. Taking the artifactual reads for K06988 Reut_B5423 from *Ralstonia* as an example, we summarized the species with the best hits in BLASTX results for the 3' flanking fragments. Up to 83% resembled to the homologs in *Cupriavidus necator* N-1 (identity 56–62%; positives 76–82%). A few of the others clearly belonged to *Methylobacterium* species because the sequences were more similar to homologs from *Methylobacterium* than from other bacteria. Therefore, chimeras of gene fragments in the artifactual reads were confirmed.

To understand the formation of the chimeras, the artifactual reads for gene K01627 in Sed222 were further studied. Based on taxa of the best hits in BLASTX search, the K01627 genes could be assigned into three species: *C. taiwanensis* (47%), *Burkholderia ambifaria* (41%), and *Variovorax paradoxus* (12%). A total of 338 artifactual reads contained chimeras, which were an integration of the homologs from *B. ambifaria*/*V. paradoxus*

and that from *C. taiwanensis*. However, the chimeric phenomenon was not widespread because the artifactual reads for the other genes K07115, K01409, and K00257 did not contain obvious chimeric gene fragments from different species.

### Variants of the stem-loop model

The folding structures of the artifactual reads and internal arrangement styles of the gene fragments were summarized; a schematic model was then proposed for individual genes. The stems for K06988 and K01627 reads along with three more under the schematic model are shown in Fig. S5. The large part of the stem for K06988 was derived from 5' of the gene region, but that for K01627 was from 3' of the gene region. At the integration position, no large unknown fragments were inserted although the similarity between the reads and the genes at these positions was low. The central stems of three other genes including K07115, K01409 and K00257 had no insertions between the translocated fragments. Moreover, the stem might be shorter because the fragments adjacent to the integration sites were sometimes shorter than 10 bp, particularly for the shaded part in Fig. S5. In case that the sequences by which the major part of the stem-loop was constructed, appeared in the flanking region, the size of the flanking sequences was generally longer to enable the conformation of the stem-loop. For the K01409 reads, the average size of the flanking sequences was 37 bp for both ends; for the K07115 reads, it was 34 bp for 5' sequences and 29 bp for 3' sequences; for the K00257 reads, it was 22 and 24 bp for 5' and 3', respectively.

### DISCUSSION

In this study, artifactual pyrosequencing reads were uncovered in MDA-amplified sediment metagenomes. They were mostly redundant gene fragments, with stable secondary structures, translocations and chimeras. The translocated fragments belonged to neighboring parts of the homologous genes. A variety of strong DNA secondary structures were displayed in the reads, allowing us to propose a stem-loop model for interaction of the fragments. However, a fraction of these short reads (<40%) could not fit the hypothetic model. This probably stemmed from the DNA extraction and pyrosequencing preparation, in which the artifactual fragments would be truncated. Furthermore, the artifactual reads for K01440 in Sed105 were likely generated by different means. The typical translocations were confirmed only in a low proportion of short reads; head-to-head translocated gene fragments in some reads could individually form a stem-loop structure. Chimeras were also observed in this study, but the formation mechanism might be different from a previous report (*Lasken & Stockwell, 2007*). In this study, the fragments in the chimeras were strictly matched to the neighboring gene regions, whereas the translocated fragments in the chimeras were separated by a genomic region of up to 10 kb in the other model.

There have been some early studies on several deep-sea sediment metagenomes (*Biddle et al., 2008*; *Biddle et al., 2011*; *Inskeep et al., 2010*; *Quaiser et al., 2011*), but a large number of short copies of gene fragments have never been reported. Researchers might have ignored the short DNA fragments in the samples, or did not use the same protocols as in this study. This study may shed some light on the process responsible for

massive generation of these artifactual reads in the sediment metagenomes. The frequently observed translocations and secondary structures in the artifactual reads are the probable cause of the artifacts. They were not ascribed to 454 pyrosequencing because the artifactual reads generated by the 454 platform were identical and translocations in the artifacts have not been observed (*Gomez-Alvarez, Teal & Schmidt, 2009*). Moreover, the artifacts introduced by the 454 could affect many more genes, instead of the small number of genes in our metagenomes. Therefore, it is highly likely that the artifacts observed in this study were the result of MDA treatment. A study suggested that small DNA fragments with complex conformation will be amplified independently during MDA (*Shoaib et al., 2008*). Considering the high abundance of the artifactual reads in the metagenomes, these translocated DNA fragments with the stable secondary structure might have been more efficiently amplified in the MDA reaction. Additionally, we also noticed many nucleotide substitutions that made more pairings in the stems than in the genes. The substitutions may not occur on the genes, because the formation of strong stem-loop structures resulted from the substitutions might prohibit transcription. Instead, the MDA amplification of the artifactual reads could probably have created the nucleotide replacements which strengthened the secondary structures. The secondary structures could have first formed in the extracellular DNA, but might also originate from intracellular genomic DNA. In which steps the translocations occurred is unknown at present, but the secondary structures might have been stabilized during the subsequent amplification.

At present, we do not yet have a satisfactory answer to the question of how the translocations happened and consequently got massively amplified. However, this study reminds us that DNA contamination, particularly extracellular DNA, in a sediment sample should be removed before MDA amplification and pyrosequencing. Otherwise, we suggested that short reads and usually abundant metagenomic reads (<220 bp in this study) should be excluded from subsequent analyses. Finally, biological significance may be explored if the abundant short copies of gene fragments with stable structures persist in the metagenomes for pure unamplified DNA samples. Finally, we did not observe abundant artifactual reads in Sed12, although MDA amplification was also applied to the raw DNA sample. This seems exceptional to the observations in other sediment layers. Since this study is based on one metagenome from each layer, whether the abundant artifactual reads will repeatedly occur in the same genes from the same layer remains a question for future research.

## ACKNOWLEDGEMENTS

We thank Dr. Paul Harrison for comments to the manuscript.

### Funding

This study was supported by the National Basic Research Program of China (973 Program, No. 2012CB417304), and awards from Deepsea Institute of Chinese Academy of Science and from the King Abdullah University of Science and Technology (SA-C0040/UK-C0016)

to P.Y. Qian. The funders had no role in study design, data collection and analysis, decision to publish, or preparation of the manuscript.

### Grant Disclosures

The following grant information was disclosed by the authors:
China 973 Program: No. 2012CB417304.
Award from Deepsea Institute of Chinese Academy of Science.
Award from the King Abdullah University of Science and Technology: SA-C0040/UK-C0016.

### Competing Interests

Pei-Yuan Qian is an Academic Editor for PeerJ.

### Author Contributions

- Yong Wang conceived and designed the experiments, performed the experiments, analyzed the data, wrote the paper.
- On On Lee performed the experiments, wrote the paper.
- Jiang Ke Yang performed the experiments, contributed reagents/materials/analysis tools.
- Tie Gang Li performed the experiments, analyzed the data, contributed reagents/materials/analysis tools.
- Pei Yuan Qian conceived and designed the experiments, wrote the paper.

### Supplemental Information

Supplemental information for this article can be found online at http://dx.doi.org/10.7717/peerj.69.

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
