# Peer review of "Artifactual pyrosequencing reads in multiple-displacement-amplified sediment metagenomes from the Red Sea"

_PeerJ, doi:10.7717/peerj.69_

## Round 0.1 · original submission · Major Revisions

The referees raised a considerable number of concerns, particularly with respect to the clarity of the manuscript. As it stands, the manuscript will require considerable overhaul to satisfy PeerJ's standards. However, the fact that both reviewers noted interesting aspects to your finding makes me think that such revision would be worthwhile for the community.

Reviewer 1 ·

Basic reporting

* Overall, the manuscript is written in substandard English which often leads to ambiguous statements. Specifically, the introduction is full of unusual language constructs and the abstract is in the need of a complete rewrite to fix the language and provide more context (e.g. define what a “gene fragment” is). I think that the article should be proofread by a native speaker or a professional before resubmission. Without trying to be comprehensive, here are some portions of the text which are in a need of improvement or clarification, with some suggestions where applicable:

- In the first three sentences of the abstract the use of past tense is unwarranted.
- Use of ambiguous phrases in the abstract: “enormous short reads”, “Richness of the short reads”, “were found to be a part of genes”, “ultrashort copies of gene fragments”.
- 65-79: unjustified use of past tense/passive voice, also typical to the rest of the manuscript.
- 69: “is thus rapidly developed”
- 70: “pyrosequencing quality”
- 71: “monopoly nucleotides” - homopolymers?
- 73: “rare environments” - low quantity of DNA?
- 73: “amplified using such as”
- 81: “bias effects” - biasing effects or just biases
- 83: “highly abundant”
- 107: “proteins were originated”
- 115: what is a “metaread”?
- 141: it should be “lower than *the* overall pyrosequencing error rate”
- 152: “to show abnormally abundant short reads”
- 161: “focusing on the genes in the short reads” - maybe “focusing on the gene of origin of the reads”
- 225: “highest similarity was corresponded”
- 228: “GC content gap” - GC content discrepancy?
- 254: “unknown fragment” - “inserted fragment” or “fragment of unknown origin” reads better
- 258: “reversely complementary” -> “reverse complement of the ...”
- 261: “They were a merging of... ”
- 284: What does it mean that the reads “were pooled”?
- 305: translocation events “were indeed popular” - “were indeed frequent”
- 341: “This may not mean...” - ambiguous sentence.
- 360: “were likely massively generated”

* The term “short reads” is used very often these days and it usually refers to data produced on the Illumina platform. I think that using a less overloaded term would improve the readability (e.g. “artifactual reads”).

* The title needs rephrasing, as “short copies of gene fragments” sounds a bit cryptic for the non-specialist. My suggestion is “Abundant artifactual reads in amplified sediment metagenomes”.

* The introduction has to be improved beyond fixing the issues of language and style. The manuscript essentially presents an artifact specific to a given protocol, but there is not enough background on the protocol and the potential biases in order to follow the rest of the article. I have the following suggestions in order to improve the introduction:
- Please provide a description of the individual steps in the protocol used and point out the potential artifacts present in these different steps.
- Please provide a more in-depth literature review on the potential artifacts of the MDA protocol. The paper by Lasken and Stockwell (http://www.biomedcentral.com/1472-6750/7/19) seems to be appropriate to cite.
- Emphasize the artifacts which can generate the translocations and fragment reassortments observed in the data.
- It would be very useful to provide a figure illustrating the protocols and the potential artifacts.
- Given enough information on the artifacts, it would be easy then to define what a “gene fragment” is.
- What are the origins of the “extracellular DNA” mentioned throughout the article?

* Regarding the figures:
- Figure 1: please color-code the lines and make them thinner to compensate for the overlaps.
- It would be very useful to color-code the “fragments” in the secondary structures.
- It should be clarified in the text what the structure shown on Figure 3C is.
- Please reconsider whether Figure 5 is worthy of inclusion in the main text.
- Supporting Figure 1 is hard to interpret as a scatterplot. It would be nicer to have heatmaps instead with “start” and “end” as axes.

Experimental design

* 107: Is it justified to assume that the length of the genes belonging to species form the same genus are the same? If yes, could you explicitly state this?

* 115: Please state what is the expected average length of a *normal* 454 read.

* 179: Is the length of the control gene K00984 similar to the gene showing the unusual read length distribution? Genes of different length can produce fragments with significantly different length distribution even when subjected to the same fragmentation process. Hence, if there are large differences in gene size, I think it would be a good idea to show by simulation that one cannot get significant differences in the length distribution under random fragmentation.

* Strangely for a study about artifacts, the experimental design lacks technical replicates. This does not affect strongly the main message of the article, but having them would make the results more conclusive.
Also for this reason it remains unclear why the Sed63 sample shows less artifacts: in the lack of replicates the cause could be small differences in the experimental conditions or different composition of the sample. This might be worth mentioning in the discussion.

* 427: How were the “pyrosequencing artifacts” detected? Please describe it in the “Materials and Methods” section.

Validity of the findings

* PeerJ policy explicitly advises the authors to deposit the data in public repositories. However, there is no SRA/ENA accession number provided for the sequencing data scrutinised in the manuscript. Please provide an accession number in the revised manuscript.

* I found the Results section to be hard to follow and crammed with details which detract from the main focus of the article. I have a couple of suggestions for improving this section:
- The gist of the article can be summarised as: MDA artifacts generate secondary structures with increased stability and replicative advantage, which can explain the presence of abundant artifactual reads. It would be nice to reorganise the results along these lines.
- Following the point above, instead of Figure 3, pick a representative gene and present its secondary structure alongside the secondary structure of the reads mapped to the region. Emphasize the rearrangements and point mutations increasing stability by color-coding and other visual means.
- I do not think that a very in-depth discussion of all the findings is necessary in the *main text*. What about summarising the impact of different artifacts (chimeras, translocations, point mutations, etc.) in a table and move the detailed per-gene descriptions to the supporting material?
- Following the point above, chimeric reads and translocations should be viewed as mechanisms generating potentially more stable structures. Their description should be more concise.

* 191: “implying abundance of sequence variants”. Please clarify this statement and the possible nature of these variants.

* 224: Can the drop in similarity explained by the mutations leading to more stable structures? If yes, this should be mentioned in the discussion.

* 228: Is there any clue about the cause of the GC content discrepancy? Can this be caused by an artefact of the MDA amplification (similarly to PCR preferring GC content around 50%) or is more likely attributable to the biological source of the reads? Does this have any significance? If not, then maybe it should be removed to keep the text more focused.

* 245: “but their presence in BLASTX results could not be interpreted” - this suggests, that there is a clear-cut interpretation for the K01627 gene, however this should be made more explicit, as for the naive eye there are no *obvious* and *striking* differences in the contexts of the hotspots.

* 247: “The short reads were associated with the alignment hotspots were checked for the secondary structure” - it is not clear if this refers to reads from all genes or reads from one particular gene.

* On line 306, the so-called “universal model” is mentioned for the first time, but it is not defined properly anywhere in the text. Also the section “Variants of the universal model” should be shortened or moved to the supporting material.

* 314: it is not clear what the “pairwise distances” refer to.

* The conclusion of the article is that the artifactual reads are produced during MDA, yet at line 433 there is an advice to remove small fragments *before* amplification!

Additional comments

The authors report the presence of abundant artifactual reads in a sediment metagenomic data produced using a whole genome amplification protocol and 454 sequencing. The observation is interesting, however the manuscript has to be significantly improved in terms of style and readability. Specific points are elaborated in the other sections.

·

Basic reporting

This manuscript would benefit from two major revisions which would dramatically improve its quality:

1) The Authors are advised to secure the assistance of a native english speaker for correction of their text. At present, the english is difficult to penetrate to get at the science underneath.

2) The bioinformatic pipeline used to analyze the results is a complicated, multi-step process and would be greatly clarified by the inclusion of a Figure depicting the methods in a flow diagram.

The authors describe the use of 'Whole Genome Amplification' in metagenomic samples, when I think what they are actually referring to (as per the included reference Dean et al 2002) is known as multiple displacement amplification (MDA). Whole Genome amplification implies the use of single amplified genomes, which they are not using in this study.

Experimental design

I hope that the concerns raised here are due, at least in part, to my misunderstanding of the manuscript that will be improved once advice is sought on the english used. At present, the manuscript appears to suggest surprise that many of the short reads align to the same parts of the genes identified in this study. However, this stochastic and systematic bias of amplification using MDA has long been established. Indeed, the hyperbranching of DNA that occurs using phi 29 polymerase almost guarantees that parts of genes will be more amplified than others, which is why techniques such as digital normalization (Brown et al. 2012) of MDA-amplified genomes superbly assists in assembly.

Similarly to this, the expectation that the gene fragments in the reads would be uniform (line 199) is unlikely as a uniform distribution would require independence of each of the reads. However, MDA works precisely because the reads generated are non-independent. It is therefore more likely that one would see a gamma or negative-binomial distribution of reads rather than a uniform distribution.

It is interesting that the abundance of short fragments is not seen in all datasets and I wonder whether this is just due to the probabilities of sampling (i.e. what is the false-negative rate) or whether it is a function of the amount of DNA in the starting preparation prior to amplification. As DNA amounts decrease towards 0, the repetition of DNA will increase assuming an equal concentration of primers.

Validity of the findings

I feel uncomfortable judging whether or not the findings in this paper are statistically robust and/or valid until I see a future draft where the language describing the methods and discussion is more clear.

---

## Round 0.2 · Major Revisions

This revised manuscript is a considerable improvement over the initial submission. However, a number of issues remain, mainly regarding the clarity/writing of the manuscript, but also a few deeper issues.

Importantly, I would like to draw your attention to PeerJ's policy regarding data deposition (https://peerj.com/about/policies-and-procedures/), which requires that you make the data available alongside publication. Reviewer 2 provides a concrete suggestion of how you could deal with your particular situation.

Both referees have gone out of their way to provide detailed, helpful feedback on the basic reporting and writing aspects of the manuscript. I invite you to address their feedback in earnest, possibly soliciting the assistance of a professional editor or a native English-speaking colleague.

·

Basic reporting

This paper highlights the fact that when MDA is used to amplify DNA from sediment metagenomes three observations are made:

1) Many of the sequenced reads are short
2) The short reads appear to derive from a narrow range of locations on a very narrow range of genes.
3) Many of these short genes appear to have a stable secondary structure.

The conclusion is therefore reached that a reasonable hypothesis is that this secondary structure is causal to the high replication of these sections during MDA.

The arguments made are convincing that there is strong evidence for (1), (2) and (3) and the authors go to great lengths to analyse the data in different ways to prove the point.

Interestingly, in all post-sequencing pipelines I've seen used, the first step is to throw away short sequences once they have been trimmed by quality score. With 454 GS-FLX titanium, this would typically be reads <300bp and would thus negate the problems highlighted in this paper. However, it is interesting that a possible cause for these short, low quality reads has been identified

Experimental design

No Comments (see general comments)

Validity of the findings

No Comments (see general comments)

Additional comments

Line 92: Should read: 'undesired background amplification may occur and occasionally occupy'
Line 99: Brief explanation of how Pan resolved this issue would be helpful.
Line 119: What was the concentration of DNA from the extracted samples? The Nextera XT sequencing platform from Illumina can be run using 1 ng of DNA. It would be useful to know if it wasn't enough even for this.
Line 145: Version of BLASTX used should be stated
Line 152: When looking to see if proteins in the best BLAST hits belonged to the same genera, was a cutoff used? Top 10? Top 20?
Line 157: Might be worth mentioning here that the cutoff of 220bp was derived from the distribution shown in Figure 2.
Line 159: The meaning of the sentence starting 'When BLASTX results showed...' remains unclear to me.
Line 163: What would make conversion to positive-strand sequences necessary?
Line 166: Why the switch from ClustalW to MUSCLE? Should also read v3.6.

Results paragraph starting at line 191: It would be useful to include a notch-plot of the read length distributions to show that there is significant difference between the samples and to display the distribution. Also, I would be interested to know what the quality scores were for the sequences <220 bp. Was there a significant decrease in the quality of these reads?

Line 206-212: I would leave out the description of what these KEGG genes encode. It is not relevant to the story and is based upon an assignment of function from a very short piece of DNA.

Line 216: The bulge in Fig 2. appears to be between 160-200bp for Sed183, not 50-160bp

Line 253: Figure S1 would be clearer if it were plotted as an abundance plot at each position, rather than relying on the reader to infer density from the number of circles at each position.

Line 290: should read 'BLASTX results'.

Line 326: should read '272 5' flanking sequences' (no comma required)
Line 331: should read 'a total of 937 reads were'
Line 337: should read 'hotspots were not taken'
Line 347: Was this a best-hit search against KEGG again?
Line 356: 'chimeras' is spelled incorrectly
Line 385: The sentence 'This probably stemmed...' needs to be made clearer as to how this would account for 40% of the reads not fitting the hypothetical model
Line 400: should read 'artifactual reads'
Line 403: either 'highly similar' or 'identical'.

Reviewer 2 ·

Basic reporting

The presentation of the results and the style of the manuscript has significantly improved compared to the initial submission (but see section #3). The introductory section is now more readable and informative. However, some key issues remain unsolved and some new issues have been introduced and I cannot recommend the manuscript for publication before they are properly addressed.

1# Issues carried over from the previous round:

* Please give an explicit definition of what do you mean by “gene fragments” early in the manuscript, preferably in the introduction.

* Please explicitly denote in the text what do you mean by “the universal model” in the section “Stable secondary structures in genes and artifactual reads”, or earlier.

* Regarding Supporting figure 1: the authors response was: “x axis is labeled as individual reads. For each read, the start and end positions were shown in y axis. The figure is not a scatterplot.” - I understand that supporting figure 1 is not a simple “scatter plot”, however it is not an intuitive representation of the read alignments. I suggests that the authors consider an alternative way to represent the alignments (e.g. showing the alignment pileup itself).

2# Points needing further clarification:

* 184: “lower than the overall pyrosequencing error rate” - I guess the correct phrase would be “lower than the threshold suggested by the pyrosequencing error rate”

* 200: “Thus reads shorter than 220bp might contain artifactual pyrosequencing reads.” - This statement feels slightly circular, as the artefactual reads are defined as abundant reads shorter than 220bp.

* 231: To the superficial reader 6180 reads assigned to 1189 clusters night not suggest a significant redundancy. Are these numbers right? If yes, then this portion needs some more clarification.

* 245: Please clarify “Thus distribution of gene fragments in the reads dramatically differed from that of all the corresponding metagenomic reads.”

* 413: Does “instability of the genome” mean some specific process or it is just equivalent to mutations having a large deleterious effect. If the authors refer to a specific process, then it would be nice to cite a relevant paper.

* 309: “were found with four Gs originally on the modified genes” - please rephrase this

* 390: “both fragments could form a stem-loop structure” - please clarify this

* 423: “Otherwise, those short reads...” - please rephrase this sentence. I suggest “Otherwise, we suggest that short and unusually abundant metagenomic reads (<220bp in this study) should be excluded from subsequent analyses.”

3# Suggestions to improve language and style:

In general, the style of the manuscript could still use some improvement (excluding the introduction).

* 39: suggested alternative to “Translocated gene fragments...”: “Gene fragments translocated from neighboring regions were identified in these reads.”
* 41: “The gene fragments in these reads...” -> “Reads containing these gene fragments could form a strong stem-loop structure.” - if appropriate
* 42: “reads could fit the structural model” -> “structural models” - there are more structures even for a single gene.
* 82: “direct accession of genomes” - does this mean “direct assessment of genomes”?
* 130: “They were regarded as an artifact...” -> “Hence we conclude that the observed abundant short reads are artefacts of the MDA treatment.”
* 144: (KEGG)(link)(v51) -> “(KEGG, link, v51)”
* 203: Please rephrase “Artifactual reads could be identified by focusing on the genes in the short reads.” - the meaning is not clear.
* 206: “over-represented by the short reads” -> “over-represented in the short reads”
* 255: “Then, most...” -> “Most...”
* 281: “characteristics of the read alignment” -> “alignments”
* 303: “integrated with” -> “integrated within”
* 356: “chimaras” -> “chimeras”
* 365: “were displayed” -> “are shown”
* 383: “Variant strong DNA secondary structures” -> “A variety of strong DNA secondary structures”
* 402: “probably the cause” -> “the probable cause”
* 414: “replacements to strengthen” -> “which strengthened”
* 421: “were constructed and massively amplified” -> “happened and consequently got massively amplified”
* 427: “... was excluded from the artifactual reads” -> “Finally, we did not observe abundant artefactual reads in Sed12...“

5# Suggestions regarding the structure of the manuscript:

I believe that the following suggestions could greatly improve the readability and flow of the manuscript. However, I would like to ask the editor to decide whether is necessary to implement them.

* The subsections about “more translocations”, “chimeric fragments” and “variants of the universal model” still seem rather long to me. However, their main points are nicely summarised in lines 384-394 - I suggest to move these lines to the results section and move the subsections in the supporting text.
* 344: Remove the sentence “A question is whether...”
* 239: Remove the sentence “More details regarding the length distribution...”
* 320: Remove the sentence “The frequency of the translocations...”
* 250: Remove the sentence “Whether the artifactual reads were located at a certain gene region...”
* 266: Remove the sentence “The artifactual reads aligned to the highly covered...”

Experimental design

-

Validity of the findings

* PeerJ policy explicitly advises the authors to deposit the data in public repositories. However, there is no SRA/ENA accession number provided for the sequencing data scrutinised in the manuscript. Please provide an accession number in the revised manuscript. Response: We are preparing a new manuscript with the metagenomes. Therefore we will release them later. More issues about the data release will be discussed with our collaborates in KAUST.

I believe that referencing the accession number identifying the dataset is a minimum requirement for publication. I hope that the authors will be able to agree with their collaborators on this issue. It is possible to submit the dataset to SRA and reference the accession number in the present manuscript while keeping the dataset private for a year, which might solve the data release issue.

---

## Round 0.3 · accepted · Accept

Thank you for addressing all the concerns of the referees. I am also pleased that you have deposited all sequencing reads relevant to your study to a public repository, in compliance with PeerJ's policy.